# The role of eCIRP in bleomycin-induced pulmonary fibrosis in mice

**Siavash Bolourani[1,2,3], Ezgi Sari[1], Max Brenner[1,3,4‡]\*, Ping Wang[1,2,3,4‡]\***

**1** Center for Immunology and Inflammation, Feinstein Institutes for Medical Research, Manhasset, NY, United States of America, **2** Elmezzi Graduate School of Molecular Medicine, Manhasset, NY, United States of America, **3** Department of Surgery, Donald and Barbara Zucker School of Medicine at Hofstra/Northwell, Manhasset, NY, United States of America, **4** Department of Molecular Medicine, Donald and Barbara Zucker School of Medicine at Hofstra/Northwell, Manhasset, NY, United States of America

‡ These senior authors contributed equally to this work.
\* mbrenner@northwell.edu (MB); pwang@northwell.edu (PW)

## Abstract

### Objective and design

We examined the role of eCIRP in the pathogenesis of bleomycin-induced pulmonary fibrosis (PF).

### Material and methods

Publicly available gene expression omnibus datasets were analyzed for the expression of CIRP in lung samples from patients with PF. Wild type (WT) or CIRP$^{-/-}$ mice received daily injections of 10 μg/g bleomycin for 10 days. A subset of bleomycin-injected WT mice was treated with the eCIRP antagonist C23 (8 μg/g/day) from day 10 to day 19. At three weeks, transthoracic echocardiography was performed to measure the degree of pulmonary hypertension, and lung tissues were collected and analyzed for markers of fibrosis.

### Results

Analysis of the mRNA data of human lung samples showed a significant positive correlation between CIRP and α-smooth muscle actin (α-SMA), an important marker of fibrosis. Moreover, the expression of CIRP was higher in patients with acute exacerbation of PF than in patients with stable PF. CIRP$^{-/-}$ mice showed attenuated induction of α-SMA and collagens (Col1a1, Col3a1), reduced hydroxyproline content, decreased histological fibrosis scores, and improved pulmonary hypertension as compared to WT mice. WT mice treated with C23 also had significant attenuation of the above endpoint measure.

### Conclusions

Our study demonstrates that eCIRP plays a key role in promoting the development of PF, and blocking eCIRP with C23 can significantly attenuate this process.

**Data Availability Statement:** All relevant data are within the paper and its Supporting Information files.

**Funding:** This study was supported by the National Institutes of Health (NIH) National Heart, Lung, and

Blood Institute (NHLBI) grant R01HL076179 (PW), and National Institute of General Medical Sciences (NIGMS) grant R35GM118337 (PW). The funding agencies played no role in the study design, data collection and analysis, decision to publish, or preparation of the manuscript. NHLBI: https://www.nhlbi.nih.gov/ NIGMS: https://www.nigms.nih.gov/.

**Competing interests:** The authors have declared that no competing interests exist.

## Introduction

Pulmonary fibrosis (PF), characterized by unrestrained production and accumulation of extracellular matrix components, is a pathology common in many chronic lung diseases such as idiopathic PF (IPF), sarcoidosis, and systemic sclerosis. Chronic inflammation caused by inhalation of foreign particles and autoimmune disease often results in PF [1, 2], although a causal role for inflammation has not been determined in IPF [3]. Irrespective of its etiology, PF is associated with high morbidity and mortality [4–6].

It has been long known that inflammation plays a key role in the triggering and persistence of fibrotic processes in the lungs [7]. To this end, many inflammatory mediators such as damage-associated molecular patterns (DAMPs) and proinflammatory cytokines have been found to play a role in the exacerbation of fibrosis [8–12]. Extracellular cold-inducible RNA-binding protein (eCIRP) is a novel DAMP discovered by our lab to play a key role in sepsis, hemorrhagic shock, acute lung injury, renal and intestinal ischemia-reperfusion injuries, and ischemic stroke [13–17]. eCIRP induces these effects through its binding to the toll-like receptor 4 (TLR4) and specifically TLR4-myeloid differentiation 2 (MD2) complex [18].

While the effect of eCIRP in acute inflammation and acute lung injury has been studied, little is known about its role in chronic inflammatory diseases of the lungs, especially the development of PF. During the past decade, evidence has emerged that DAMPs play a key role in promoting exacerbation, remodeling, and silent progression of PF [8]. Furthermore, laboratory and genome-wide association studies have implicated the TLR4-MD2 complex to be a critical mediator of pulmonary fibrosis [19–22]. Hence, we decided to examine whether eCIRP plays any role in the pathogenesis of PF and whether blocking this effect by a small molecule can ameliorate the development of PF in a mouse model.

## Materials and methods

### Analysis of gene expression omnibus datasets

The expression datasets were obtained directly from the gene expression omnibus website (https://www.ncbi.nlm.nih.gov/geo) using the *getGEO* function of R (http://www.r-project.org/) 4.0.3 library called *GEOquery* [23]. After obtaining the datasets for both GSE98925 and GSE10667, they were each normalized to the median value across all samples using package *limma* workflow [24]. To correlate the normalized mRNA count of α-smooth muscle actin (α-SMA) and CIRP in gene expression data of GSE98925, *ggscatter* function of library *ggpubr* library was used by setting the correlation method to Pearson correlation, and the coefficient was obtained. To compare the difference between the normalized count of CIRP among stable IPF patients and those with acute exacerbation of IPF in gene expression data of GSE10667, an unpaired Student's t-test was used.

### Experimental animals and model of fibrosis

Wild type (WT) C57BL/6 mice (Jackson Laboratory) and CIRP$^{-/-}$ mice with C57BL/6 background originally obtained from Kumamoto University, Japan and maintained at the Feinstein Institutes for Medical Research [25] were used in this study. Ten to twelve week-old male WT and CIRP$^{-/-}$ mice ($n$ = 10 mice per group) received subcutaneous (s.c.) injections of bleomycin (Gold Biotechnology, St. Louis, MO; 10 μg/g/day, 100 μl of 1 mg/ml solution per injection of a ~20 g mouse) (identified as WT Bleomycin and CIRP$^{-/-}$ Bleomycin group) or PBS (identified as WT Control and CIRP$^{-/-}$ Control group) daily for 10 non-consecutive days (five times per week for two weeks) as described [26]. Animals were monitored daily for changes in health and behavior. The ultrasound imaging was obtained at three weeks. Animals require no special

housing and no special handling. Since pulmonary fibrosis is painless, no analgesics or anesthetics were used. Two WT Bleomycin mice developed two or more predetermined humane endpoints (minimal or absent activity, weight loss of $\geq$ 20%, hunched or recumbent posture, minimal or no response to stimuli, grimace score = 2, body condition score $\leq$ 2, and labored breathing or respiratory distress) on day 18 and were immediately euthanized and excluded from the analysis. No mice died before meeting criteria for euthanasia. The remaining mice were sacrificed on day 22, when the lungs were collected. The left lung was fixed in formalin for histological assessment; the right lung was used for qPCR (superior and middle lobes) and measurement of hydroxyproline content (inferior and post-caval lobes).

## C23 peptide

C23 peptide (GRGFSRGGGDRGYGG) was synthesized by GenScript (Piscataway, NJ). C23 is a short 15-mer peptide derived from CIRP with a high affinity to the TLR4-MD2 complex [13]. This peptide has been shown to antagonize eCIRP on TLR4-MD2 complex and attenuate sepsis, hemorrhagic shock, lung injury, and renal and intestinal ischemia-reperfusion injury [27–30]. For the WT Bleomycin+C23 group, in addition to the bleomycin injection described above, mice were also s.c. injected with C23 peptide (8 µg/g/day) as previously described [14, 28] starting on day 10 from the start of the initial bleomycin injection until day 19.

## Isolation and analysis of real-time quantitative polymerase chain reaction

At the end of the experiments, total RNA was isolated using Trizol reagent (Invitrogen, Carlsbad, CA) and reverse transcribed to cDNA using M-MLV (Moloney murine leukemia virus) reverse transcriptase (Applied Biosystems, Foster City, CA). Polymerase chain reactions (PCR) were carried out in a final volume of 10.5 µl which included 5 µl SYBR Green PCR master mix (Applied Biosystems), and 0.031 µM of both reverse and forward primer. Glyceraldehyde 3-phosphate dehydrogenase (GAPDH) mRNA was used to normalize the amplification data and fold changes were calculated in comparison with WT Control mice using the $2^{(-\Delta\Delta Ct)}$ method. The sequence of forward and reverse primers used were as follows: GAPDH; 5'-CATCACTGCCACCCAGAAGACTG-3' (forward) and 5'-ATGCCAGTGAGCTTCCCGTTCAG-3' (reverse). α-SMA; 5'-TGACCCAGATTATGTTTGA-3' (forward) and 5'-GCTGTTATAGGTGGTTTCG-3' (reverse). Collagen type 1 alpha 1 (Col1a1); 5'-GCAAGAGGCGAGAGAGGTTT-3' (forward) and 5'-GACCACGGGCACCATCTTTA-3' (reverse). Collagen type 3 alpha 1 (Col3a1); 5'-AGGCTGAAGGAAACAGCAAA-3' (forward) and 5'-TAGTCTCATTGCCTTGCGTG-3' (reverse).

## Hydroxyproline assay

Colorimetric hydroxyproline assay kit (Cambridge, UK—catalog number: ab222941) was used to measure hydroxyproline content from samples prepared from ~100 mg of lung tissue lysates from each mouse.

## Pulmonary hypertension and echocardiography

At the end of experiments, prior to the collection of lung tissue from mice, pulmonary hypertension data was obtained using transthoracic echocardiography. Prior to performing echocardiography, mice were sedated using minimal isoflurane and maintained on a heated table. Echocardiography was conducted using a 40 MHz center frequency transducer coupled to a Vevo®3100 Imaging System (Fujifilm VisualSonics, Toronto, ON, Canada). Pulmonary acceleration time (PAT) and pulmonary ejection time (PET) were obtained and used to calculate

the PAT/PET ratio, which correlates with inverse right ventricular pressure–an indicator of pulmonary hypertension [31, 32].

## Lung histology and fibrosis scores

Lung tissues were obtained at the end of the experiment and were cut into 5-μm fragments, preserved in paraffin, and later stained with hematoxylin and eosin. Hübner's modification of the Ashcroft method was used to calculate the fibrosis score [33]. Briefly, we used the following gradings: (0), normal lung; (1), isolated alveolar septa with minor fibrotic changes; (2), fibrotic changes of alveolar septa with knot-like formations; (3), continuous fibrotic walls of alveolar septa; (4), non-confluent fibrotic masses; (5), confluent fibrotic masses; (6), large continuous fibrotic masses; (7), air bubbles; (8) fibrous obliteration. The sections were reviewed under light microscopy (Nikon ECLIPSE Ti-S) at 200x magnification. For each staining section, four random sites were observed and scored; the mean score of four sites was used as the fibrosis score of each lung section. The sections were assessed by an investigator blinded to the study groups.

## Statistical analysis

Data are expressed in the bar graphs as mean and standard deviation (SD), with individual values shown by colored points. The comparisons of WT and CIRP$^{-/-}$ mice's molecular and physiological markers were performed using two-way analysis of variance (ANOVA) followed by Tukey's multiple comparison test with the factors of two-way ANOVA being CIRP gene expression status (WT vs. CIRP$^{-/-}$) and bleomycin exposure status (Bleomycin vs. Control). The comparison for those with the C23 treatment group was done via one-way ANOVA followed by Tukey's multiple comparison test. Differences between the experimental groups reaching a p-value of 0.05 or less were considered statistically significant.

## Ethics

The expression dataset comparisons used deidentified data publicly available at the gene expression omnibus website and, therefore did not require IRB approval. All experiments involving live animals were carried out in accordance with the National Institutes of Health Guide for Care and Use of Laboratory Animals [34] and were reviewed and approved by the Institutional Animal Care and Use Committee (IACUC) at the Feinstein Institutes for Medical Research.

## Results

### The positive correlation of CIRP with α-SMA, and upregulated CIRP expression in the lungs of IPF patients during exacerbation

We examined two publicly available gene expression omnibus (GEO) profiles: GSE98925, containing the array profile of microdissection from multiple fibroblastic foci from the lung tissues of 13 IPF patients either during a biopsy or after lung explant, and GSE10667, containing the array profile of whole lung tissue samples from IPF patients either with biopsy or after lung explant. This included 23 stable IPF patients (UIP) and 8 patients with acute exacerbation (AEx). We sought a correlation between the normalized mRNA count of CIRP and α-SMA levels in GSE98925 data of fibroblastic foci. The analysis showed a positive correlation between CIRP and α-SMA levels (R = 0.39, p = 0.0017) (**Fig 1**). It is important to note that the correlation held even after we removed the samples that were two standard deviations from the mean of each expression. Furthermore, we looked at the expression of CIRP mRNA in the tissue samples profile obtained from GSE10667, which showed a significant increase in the normalized CIRP mRNA in the lung samples during exacerbation (p<0.05) (**Fig 1 insert**). This

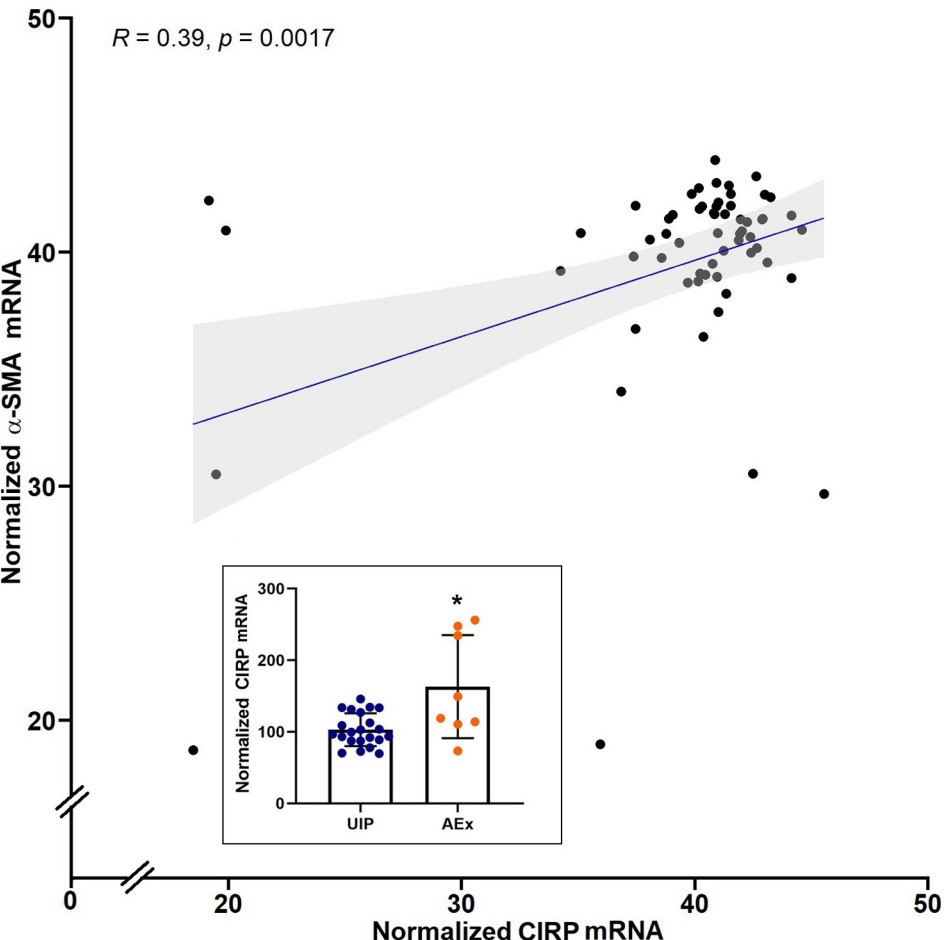

**Fig 1. CIRP mRNA expression correlates with α-SMA in IPF patients and is increased during acute exacerbation of IPF.** Examinations of two publicly available gene expression omnibus (GEO) profiles: GSE98925 and GSE10667 (insert): After obtaining mRNA expression data using the getGEO function, expressions were normalized across all samples using the package *limma* workflow. The figure shows Pearson correlation and the coefficient (R = 0.39) obtained between mRNA expression of CIRP and α-SMA, along with p-value (p = 0.0017) shown for GSE98925, which is the mRNA array profile of microdissection from multiple fibroblastic foci from the lung tissues of 13 IPF patients either from the biopsy or after lung explantation. The insert shows the difference between the normalized CIRP mRNA in stable IPF patients (UIP) vs. those with acute exacerbation of IPF (AEx) in data from GSE10667. Unpaired Student's t-test was used to determine the significance. * p<0.05 vs. UIP. The sample GSM269765 was removed due to normalized CIRP mRNA for that sample being more than 3 standard deviations away from the means.

analysis showed that not only does the expression of CIRP correlate with α-SMA in IPF patients, the levels of CIRP increases during acute exacerbation of IPF.

## Molecular markers of fibrosis are ameliorated in CIRP-/- mice

To examine the fibrotic role of eCIRP, we examined the mRNA expression of α-SMA, Col1a1, and Col3a1, and the corresponding hydroxyproline content in the lung tissue of WT Control, WT Bleomycin, CIRP$^{-/-}$ Control, and CIRP$^{-/-}$ Bleomycin mice. Our analysis showed that there was a 10.65- and 4.27-fold increase in α-SMA and Col3a1 expressions respectively in WT mice when comparing Bleomycin vs. Control group (p = 0.0180 for α-SMA mRNA p = 0.0002 for Col3a1 mRNA), however, there was no significant increase seen in CIRP$^{-/-}$ mice. Furthermore, mRNA expressions of α-SMA and Col3a1 were attenuated by 89% and 64% (p = 0.0030 for α-

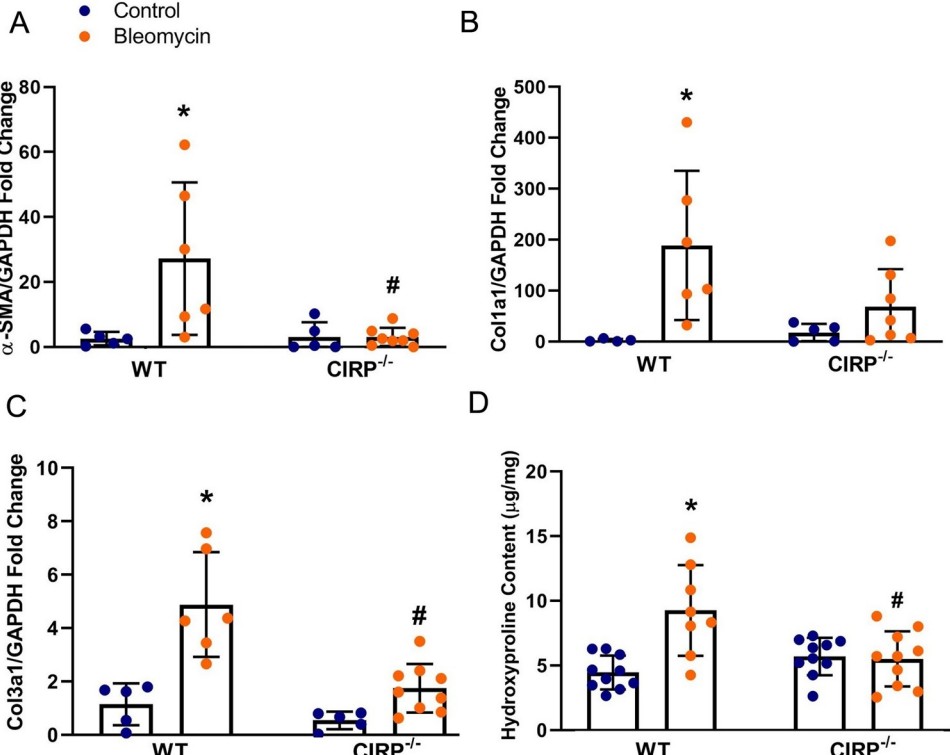

**Fig 2. Deficiency in CIRP attenuates molecular markers of lung fibrosis in mice exposed to bleomycin.** For α-SMA (**A**), Col1a1 (**B**), and Col3a1 (**C**) mRNA expression data, total RNA was isolated using Trizol reagent and reverse transcribed to cDNA using M-MLV (Moloney murine leukemia virus) reverse transcriptase. Glyceraldehyde 3-phosphate dehydrogenase (GAPDH) RNA was used to normalize the amplification data and fold changes were calculated in comparison with WT Control mice using the $2^{(-\Delta\Delta Ct)}$ method. For the hydroxyproline assay, a calorimetric Hydroxyproline assay kit was used to measure hydroxyproline content from samples prepared from ~100 mg of lung tissue lysates from each mouse (**D**). * $p < 0.05$ vs. Wild Type (WT) Control, # $p < 0.05$ vs. WT Bleomycin. Data were analyzed using a two-way analysis of variance (ANOVA) followed by Tukey's multiple comparison test with factors of two-way ANOVA being CIRP gene expression status (WT vs. CIRP⁻/⁻) and bleomycin exposure status (Bleomycin vs. Control).

SMA, p = 0.003 for Col3a1) in CIRP⁻/⁻ Bleomycin mice compared to WT Bleomycin (**Fig 2A, 2C**). There was a 72.24-fold increase (p = 0.0208) seen in Col1a1 mRNA expression of WT mice when comparing Control with the Bleomycin group, however, no significant increase was seen in CIRP⁻/⁻ mice. While Col1a1 mRNA expression level was not significantly different between CIRP⁻/⁻ Bleomycin mice compared to WT Bleomycin (due to large standard deviation of Col1a1 mRNA expression in Bleomycin WT mice), there was a 64% decrease in the mean of Col1a1 mRNA expression in CIRP⁻/⁻ Bleomycin mice (**Fig 2B**). Additionally, there was a 2.08-fold increase (p = 0.0003) seen in hydroxyproline content of WT mice when comparing Control with Bleomycin group and was not seen in CIRP⁻/⁻ mice. The hydroxyproline content CIRP⁻/⁻ Bleomycin group was attenuated by 40% (p = 0.0049) as compared to WT Bleomycin mice (**Fig 2D**).

## Physiological and histological markers of fibrosis are ameliorated in CIRP-/- mice

We examined the PAT/PET ratio (a surrogate for pulmonary artery flow) using echocardiography in WT Control, WT Bleomycin, CIRP⁻/⁻ Control, and CIRP⁻/⁻ Bleomycin mice. Our analysis showed a decrease of 32.5% (p<0.0001) in the PAT/PET ratio of WT mice when

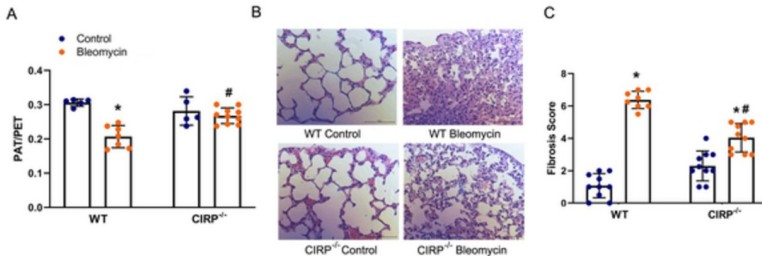

**Fig 3. Deficiency in CIRP ameliorates physiological markers of fibrosis in mice exposed to bleomycin.** Pulmonary acceleration time (PAT)/pulmonary ejection time (PET) ratio (**A**) was obtained which correlates with inverse right ventricular pressure (a surrogate for pulmonary hypertension). 5-μm lung tissue fragments preserved in paraffin, and later stained with hematoxylin and eosin (**B**) were evaluated using Hübner's modification of the Ashcroft fibrosis score (**C**). Four random sites were observed and scored under light microscopy at 200x magnification; the mean score of four sites was used as the fibrosis score of each lung section. Representative samples for the lung sections are shown at 400x from Wild Type (WT) Control, WT Bleomycin, CIRP$^{-/-}$ Control, and CIRP$^{-/-}$ Bleomycin mice (**B**). $^*$ $p < 0.05$ vs. WT Control, $^{\#}$ $p < 0.05$ vs. WT Bleomycin.

comparing the Bleomycin vs. Control group. Furthermore, there was an increase of 30% ($p = 0.0011$) seen in the PAT/PET ratio when we compared WT Bleomycin with the CIRP$^{-/-}$ Bleomycin group (**Fig 3A**). When comparing the histological effects (**Fig 3B**) there was a 5.93-fold ($p < 0.0001$) increase in the fibrosis score of WT mice when we compared the Bleomycin and Control group, while this increase was only 1.76-fold ($p = 0.0001$) in CIRP$^{-/-}$ mice. Furthermore, there was a significant decrease of 37% ($p < 0.0001$) in the fibrosis score when comparing CIRP$^{-/-}$ Bleomycin compared to WT Bleomycin mice (**Fig 3C**).

## eCIRP antagonist C23 attenuates lung fibrosis

In order to determine the effect of eCIRP antagonist C23 on bleomycin-treated WT mice, we examined the mRNA expression of α-SMA, Col1a1, and Col3a1, and hydroxyproline content in the lung tissues. Our analysis showed that while there was a 10.65-, 72.24-, 4.27-, and 2.08-fold increases in mRNA expression of α-SMA, Col1a1, and Col3a1, and hydroxyproline content respectively when comparing WT Bleomycin and WT Control group (as seen in **Fig 2**), these increases were not seen when we compared WT Bleomycin+C23 to WT Control mice. Furthermore, α-SMA, Col1a1, and Col3a1 mRNA expressions, and hydroxyproline content were decreased by 96.5, 97.4, 57.8, and 54.6% ($p = 0.0277$ for α-SMA, $p = 0.0190$ for Col1a1, and $p = 0.0046$ for Col3a1 mRNA and $p = 0.0002$ for hydroxyproline content) when we compared WT Bleomycin+C23 to WT Bleomycin (**Figs 4A–4D**).

Our analysis showed that while there was a 32.5% decrease in PAT/PET ratio when comparing WT Bleomycin to WT Control mice (as seen in **Fig 3A**), this decrease was not seen when comparing WT Bleomycin+C23 with WT Control mice. Furthermore, there was an increase of 45% ($p < 0.0001$) in the PAT/PET ratio when we compared WT Bleomycin+C23 with WT Bleomycin mice (**Fig 5A**). When we looked at the histological effects of the C23 treatment, there was a 5.93-fold ($p < 0.0001$) increase in fibrosis score when we compared WT Bleomycin with WT Control (as seen in **Fig 3C**), this increase was only 3.44 fold ($p < 0.0001$) when we compared WT WT Bleomycin+C23 with WT Control (**Fig 5B**). Furthermore, we saw an attenuation of 42% ($p < 0.0001$) when we compared concurrent C23 treatment as compared to those with bleomycin alone (WT Bleomycin+C23 vs. Bleomycin).

## Discussion

Pulmonary fibrosis is a devastating sequela of many chronic inflammatory diseases characterized by a progressive decline in lung volume capacity and resulting in high mortality [35, 36].

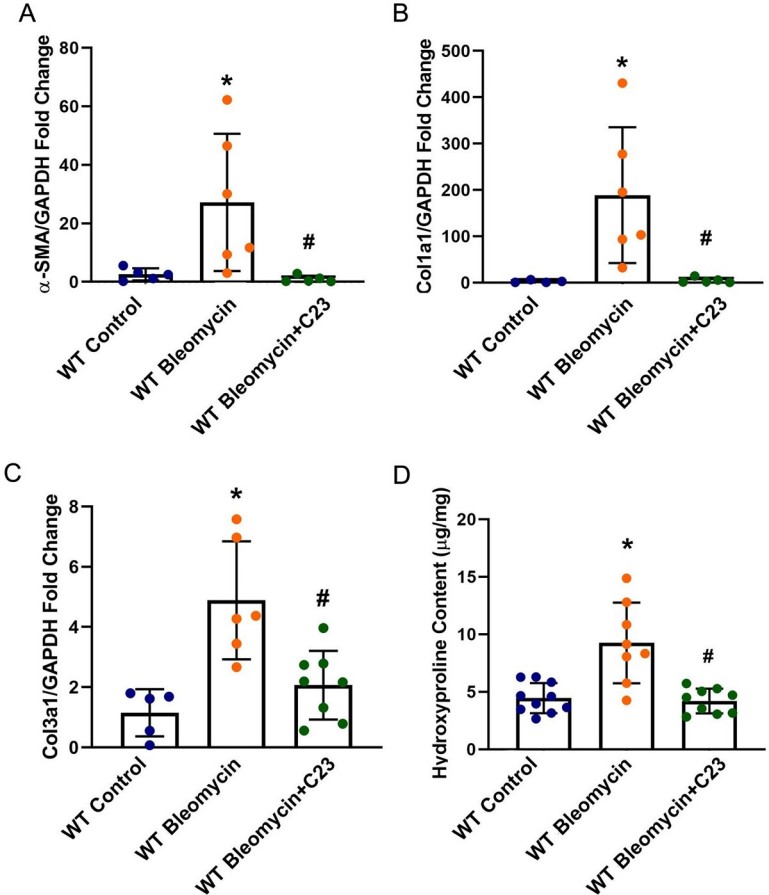

**Fig 4. The eCIRP antagonist C23 ameliorates molecular markers of fibrosis in mice exposed to bleomycin.** For α-SMA (**A**), Col1a1 (**B**), and Col3a1 (**C**) mRNA expression data, total RNA was isolated using Trizol reagent and reverse transcribed to cDNA using M-MLV (Moloney murine leukemia virus) reverse transcriptase. Glyceraldehyde 3-phosphate dehydrogenase (GAPDH) RNA was used to normalize the amplification data and fold changes were calculated in comparison with WT Control mice, using the $2^{(-\Delta\Delta Ct)}$ method. For the hydroxyproline assay, the calorimetric Hydroxyproline assay was used to measure hydroxyproline content (**D**) from samples prepared from ~100 mg of lung tissue lysates from each mouse. * $p < 0.05$ vs. Wild Type (WT) Control, # $p < 0.05$ vs. WT Bleomycin. Data were analyzed using one-way analysis of variance (ANOVA) followed by Tukey's multiple comparison test. WT Bleomycin and WT Control groups are identical to that of Fig 2.

Currently, the few therapeutics developed to ameliorate PF only modestly alter its rapid progression [37]. Estimated mortality from pulmonary fibrosis is between 46.8 to 102.6 deaths per million and there has been an increase in recent decades [38, 39]. The elevated morbidity and mortality associated with PF underline the urgent need for more effective therapeutic approaches. Studies on the role of immune cells in PF indicate that while immune cells are prominent in the early phases of the fibrotic process, the factors released by these cells within the microenvironment have a long-lasting effect [40–42]. Specifically, uncontrolled activation of macrophages and fibroblasts by DAMPs starts an inflammatory cascade that further damages pulmonary tissues, causing the further release of more DAMPs in the tissue microenvironment creating a vicious cycle that culminates in fibrosis. DAMPs exert their effect on macrophages and fibroblasts mainly through TLRs [43]. The seminal work of Bhattacharyya et al. further specified that the effects attributed to PF are largely mediated through the TLR4-MD2 pathway [20, 42, 44].

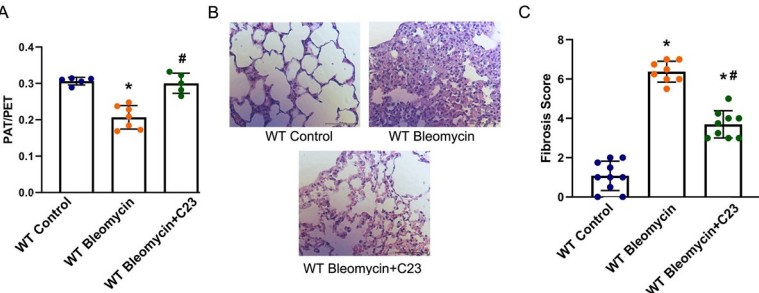

**Fig 5. The eCIRP antagonist C23 ameliorates physiological markers of fibrosis in mice exposed to bleomycin.**
PAT/PET ratio (**A**) was calculated which correlates with inverse right ventricular pressure (a surrogate for pulmonary hypertension). 5-μm lung tissue fragments preserved in paraffin and later stained with hematoxylin and eosin (**B**) were evaluated using Hübner's modification of the Ashcroft fibrosis score (**C**). Four random sites were observed and scored under light microscopy at 200x magnification; the mean score of four random sites was used as the fibrosis score of each lung section. The representative sample for the lung sections is shown at 400x from Wild Type (WT) Control, WT Bleomycin, and WT Bleomycin+C23 mice (**B**). * $p < 0.05$ vs. WT Control, # $p < 0.05$ vs. WT Bleomycin. WT Bleomycin and WT Control groups are identical to that of Fig 3.

One way to mitigate the effects of the fibrosis vicious cycle described above is to stop DAMPs with high affinity and efficacy in activating the TLR4-MD2 complex. Using surface plasmon resonance (SPR) analysis, our lab had previously shown that CIRP binds to the TLR4-MD2 complex, as well as to TLR4 and MD2 individually, with high affinity, suggesting that blocking eCIRP can ameliorate TLR4-MD2 complex signal activation [13]. Indeed, our investigation on the GEO GSE98925 mRNA profiles of samples from IPF patients' fibroblastic foci showed that there is a correlation between the expression of CIRP and α-SMA. The progression of PF is characterized at the cellular level by the differentiation of fibroblastic cells into myofibroblasts, and α-SMA is a critical marker of myofibroblast differentiation [45]. Moreover, our analysis of GEO GSE10667 showed that the expression of CIRP is particularly elevated during exacerbation of IPF in lung tissue (**Fig 1**). These findings are in line with the proposition that repetitive injury caused by acute exacerbation worsens PF not only in IPF patients but also in other chronic diseases affecting the lung [46–52]. Furthermore, in spite the lack of evidence for a causative effect of inflammation in IPF, a recent study has shown that IPF patients with elevated serum eCIRP levels had increased disease progression and all-cause mortality [53]. Taken together, these clinical observations lend support to further investigating eCIRP in a mouse model of PF as a potential target in ameliorating the fibrotic process.

The release of proinflammatory cytokines like TNF-α, IL-1β, and IL-6 in the lung tissue microenvironment by stimulated macrophages and fibroblasts contributes to the persistence of lung fibrosis by activating fibroblasts in an autocrine/paracrine fashion [10, 54–58]. Our lab has previously shown that eCIRP is a potent inducer of TNF-α, IL-1β, and IL-6 in macrophages in a TLR4 dependent manner, and that CIRP$^{-/-}$ mice have an ameliorated response to the acute injury brought on by the storm of these proinflammatory cytokines [13, 15, 18, 59]. In our recently published observations, we have seen that eCIRP is a powerful inducer of these cytokines in pulmonary fibroblasts as well [60]. Therefore, we conjectured that eCIRP can promote the persistent autocrine/paracrine activation of fibroblasts by inducing the release of proinflammatory cytokines from macrophages and fibroblasts. We focused our attention on CIRP$^{-/-}$ mice to determine if there is an attenuated response to the chronic inflammatory models of murine PF.

Rampant collagen deposition, marked by an increase in expression of Col1a1 and Col3a1, is the hallmark of fibrosis in the lung tissue extracellular microenvironment [61–64]. In addition, hydroxyproline, a major component of all types of fibrillar collagen, correlates well with the

amount of stable collagen in the lung tissue and is widely used to assess the degree of PF [65, 66]. One of the devastating complications of chronic diseases causing fibrosis in the lungs is the development of pulmonary hypertension [67–69]. We measured pulmonary hypertension by PAT/PET ratio, which is one of the most accurate non-invasive surrogates for pulmonary artery flow correlating with the inverse of pulmonary hypertension [31, 32]. As postulated, CIRP$^{-/-}$ mice lung showed ameliorated fibrotic response to bleomycin-induced PF shown by a reduction in mRNA expression of α-SMA, Col1a1, and Col3a1, and hydroxyproline content levels in the lung tissue as compared to WT mice (**Fig 2**). We also showed that this effect is not limited to molecular markers as the physiologic impact of PF is also attenuated in CIRP$^{-/-}$ mice by having an ameliorated pulmonary hypertension, and fibrosis score (**Fig 3**).

The inflammatory effect of eCIRP is caused in an extracellular milieu partially through a TLR4-MD2 complex [18] and we have shown that the CIRP-derived peptide C23 acts as a competitive antagonist inhibiting the binding of eCIRP to the TLR4-MD2 complex [13]. Since we have demonstrated that treatment with C23 attenuates inflammation and ameliorates clinically relevant endpoints in various animal models shown to be aggravated by eCIRP [28, 30, 70], we ventured to see if blocking eCIRP using C23 peptide on TLR4-MD2 complex can ameliorate the bleomycin-induced fibrotic response in the lung. We have already shown that the inflammatory pathways and cytokines are at their peak of expression on day 14 after the start of bleomycin injection and there is a drop in expression of these pathways and cytokines on day 21 [60]. This is in line with Raventti et al., who showed there is an increase in infiltration of total white blood cells, neutrophils, and macrophages in lung tissue at day 14 of bleomycin injection that decreases on day 21 [71]. Therefore, we hypothesized that by blocking eCIRP around this day, we can reduce the inflammatory phase of bleomycin-induced lung fibrosis and ameliorate the fibrotic response. Indeed, our results show that molecular and physiological markers of PF can be ameliorated in the murine model of bleomycin-induced PF by treatment of C23 from day 10 to day 19 after the first injection of bleomycin (**Figs 4 and 5**). Not much is known about C23 other than its sequence, the affinity of its binding to TLR4, and its efficacy in reducing eCIRP-mediated inflammation. Future studies should focus on C23's pharmacokinetics, immunogenicity, and possible pharmacotoxicity, as well as its potential administration by inhalation.

However, we realize this study has some limitations. The bleomycin model of PF is widely used for studies of the pathogenesis of the fibrotic process and of new treatments for PF, but it is not adequate for investigating the etiology of PF in humans. While we have shown that blocking eCIRP's effect on the TLR4-MD2 complex attenuates the fibrotic effect of bleomycin, we have not directly assessed whether this comes as a result of blocking the inflammatory phase of bleomycin-induced pulmonary fibrosis as speculated. While studies using CIRP$^{-/-}$ mice cannot determine whether the attenuated PF phenotype is due to the intra- or extracellular effects of CIRP, we have shown that blocking eCIRP with C23 ameliorates the bleomycin-induced PF, thus demonstrating the relevance of the extracellular effects of eCIRP. However, given that the injections were not done in parallel between the groups, there is a potential for a cage effect that can introduce bias into our comparisons. While we have evaluated mRNA expression of CIRP in IPF patients in publicly available GEO datasets, we could not measure eCIRP protein level in the lung tissue interstitial fluid. Nonetheless, our previous studies showed a strong correlation between increased mRNA expression of CIRP and increased eCIRP protein in serum and cell culture supernatants [13, 72–74]. Future studies obtaining eCIRP levels in the lung tissue and bronchoalveolar lavage fluid of patients with pulmonary fibrosis during exacerbations should provide further evidence that the contribution of CIRP to pulmonary fibrosis is at the extracellular level.

In conclusion, we here demonstrate for the first time that CIRP plays a crucial role in PF. We demonstrate that eCIRP contributes to PF through its effect in the extracellular microenvironment. Furthermore, we showed that the eCIRP competitive antagonist C23 can ameliorate the fibrotic response. Although the precise molecular mechanism by which eCIRP contributes to PF remains to be elucidated, we hypothesize that it is likely by dampening the inflammatory phase of fibrotic response in tissue macrophages and fibroblasts, which results in the halting of autocrine/paracrine activation to myofibroblasts. Therapeutically targeting eCIRP not only shows promise in the treatment of exacerbations of PF in chronic inflammatory diseases of the lung but also attenuating the overall progression of PF.

## Supporting information

**S1 Table. Fig 1 data and statistical analysis.** This table contains the data and statistical analysis used in Fig 1.
(XLSX)

**S2 Table. Fig 2 data and statistical analysis.** This table contains the data and statistical analysis used in Fig 2.
(XLSX)

**S3 Table. Fig 3 data and statistical analysis.** This table contains the data and statistical analysis used in Fig 3.
(XLSX)

**S4 Table. Fig 4 data and statistical analysis.** This table contains the data and statistical analysis used in Fig 4.
(XLSX)

**S5 Table. Fig 5 data and statistical analysis.** This table contains the data and statistical analysis used in Fig 5.
(XLSX)

## Acknowledgments

We would like to thank Michael Diao, MD PhD and Serdar Akkol, MD for their assistance and Monowar Aziz, PhD, for valuable discussions.

## Author Contributions

**Conceptualization:** Max Brenner, Ping Wang.

**Formal analysis:** Siavash Bolourani, Ezgi Sari.

**Funding acquisition:** Ping Wang.

**Investigation:** Siavash Bolourani, Ezgi Sari.

**Methodology:** Siavash Bolourani, Max Brenner.

**Project administration:** Max Brenner, Ping Wang.

**Supervision:** Max Brenner.

**Visualization:** Siavash Bolourani, Ezgi Sari.

**Writing – original draft:** Siavash Bolourani.

**Writing – review & editing:** Max Brenner, Ping Wang.

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
