## [Decision Letter · Decision Letter 0]

16 Feb 2022

PONE-D-21-22312

The Role of eCIRP in Bleomycin-Induced Pulmonary Fibrosis in Mice

PLOS ONE

Dear Dr. Brenner

Thank you for submitting your manuscript to PLOS ONE. After careful consideration, we feel that it has merit but does not fully meet PLOS ONE’s publication criteria as it currently stands. Therefore, we invite you to submit a revised version of the manuscript that addresses the points raised during the review process.

We look forward to receiving your revised manuscript.

Kind regards,

Manjula Karpurapu

Academic Editor

PLOS ONE

https://journals.plos.org/plosone/s/file?id=ba62/PLOSOne_formatting_sample_title_authors_affiliations.pdf".

2. In your Methods section, please provide additional details regarding bleomycin and C23 peptide used in your study and ensure you have described the source. For more information regarding PLOS' policy on materials sharing and reporting, see https://journals.plos.org/plosone/s/materials-and-software-sharing#loc-sharing-materials.

Reviewers' comments:

Reviewer's Responses to Questions

**Comments to the Author**

1. Is the manuscript technically sound, and do the data support the conclusions?

Reviewer #1: Partly

Reviewer #2: Yes

2. Has the statistical analysis been performed appropriately and rigorously? 

Reviewer #1: Yes

Reviewer #2: Yes

3. Have the authors made all data underlying the findings in their manuscript fully available?

Reviewer #1: Yes

Reviewer #2: Yes

4. Is the manuscript presented in an intelligible fashion and written in standard English?

Reviewer #1: Yes

Reviewer #2: Yes

5. Review Comments to the Author

Reviewer #1: This is an interesting and well-designed combination of human data and murine work. The authors show that eCIRP contributes to pathological manifestations of bleomycin-induced pulmonary fibrosis in mice and that small molecule blockade attenuates bleomycin effects.

The following is needed:

1) Clarify the established finding that no clear causal role is known for active inflammation in IPF.

2) Remove discussion of Covid-19 unless it is directly relevant here.

3) State that the assessment of fibrosis in mice was performed by persons blinded to the treatment/conditions. If not, then this assessment should be performed again in blinded fashion.

4) Make it very clear that studies in this mouse model cannot be used to make conclusions about human PF etiopathogenesis.

Minor: In "Materials and Methods" The expression datasets were was obtained - please fix: The expression datasets were obtained

Bottom of page 6: Ten to twelve weeks-old male - please fix: Ten to twelve week-old male

Page 11: that not only the expression of CIRP correlates - please fix: that not only does the expression of CIRP correlate

Reviewer #2: This paper is very straight forward and easy to follow. Some minor revisions include:

1. There is a duplicate sentence in the abstract. Please edit.

2. Clarify the results section of the abstract, expression of CIRP increased during exacerbation of PF in human lung tissues. Unless you also tracked CIRP expression in mice over the course of bleo treatment and not just at day 22.

3. Clarify in methods what was harvested for qPCR, mouse lung and what lobes since portions were taken for histology and hydroxyproline assays?

4. Did you notice any heterogeneity in the histological sections of mouse lung tissue? Could this variability account for the wide spread in your WT Bleo mice? Could you speak to why you might see such variability in these samples? Also was the scoring of the fibrotic lung tissue blinded?

5. I would be curious to see if preemptive treatment or co-treatment with C23 with the bleo could further reduce or prevent the development of the fibrotic phenotype observed.

6. I recommend including a section in the discussion about the application of C23 in patient treatment, potential issues, clinical use etc. to frame what is known and still needs to be understood.

The authors clearly defined their study, did not overreach their claims, and did a nice job of laying out the limitations and next steps for this line of work.

6. PLOS authors have the option to publish the peer review history of their article (what does this mean?). If published, this will include your full peer review and any attached files.

Reviewer #1: No

Reviewer #2: No

---

## [Author Response · Author response to Decision Letter 0]

28 Feb 2022

Reviewer #1: 

1) Clarify the established finding that no clear causal role is known for active inflammation in IPF.

- Thank you for this suggestion. As recommended, we have included statements in the Introduction (line 54) and in the Discussion (lines 335-337) sections indicating that no clear causal role is known for active inflammation in IPF. 

2) Remove discussion of Covid-19 unless it is directly relevant here.

- The discussion related to Covid-19 has been deleted from the revised manuscript.

3) State that the assessment of fibrosis in mice was performed by persons blinded to the treatment/conditions. If not, then this assessment should be performed again in blinded fashion.

- The histological assessment was performed in a blinded fashion. This information has been added to the Materials and Methods section (lines 153-154).

4) Make it very clear that studies in this mouse model cannot be used to make conclusions about human PF etiopathogenesis.

- We have added a sentence clearly indicating that the bleomycin mouse model is not adequate for investigating the etiology of PF in humans (Discussion, lines 389-391).

Minor: In "Materials and Methods" The expression datasets were was obtained - please fix: The expression datasets were obtained. 

- The change has been made (line 76).

Bottom of page 6: Ten to twelve weeks-old male - please fix: Ten to twelve week-old male

- The change has been made (line 89).

Page 11: that not only the expression of CIRP correlates - please fix: that not only does the expression of CIRP correlate

- The change has been made. Thank you! (line 187).

Reviewer #2: 

1. There is a duplicate sentence in the abstract. Please edit.

- Thank you for identifying this duplication! The change has been made.

2. Clarify the results section of the abstract, expression of CIRP increased during exacerbation of PF in human lung tissues. Unless you also tracked CIRP expression in mice over the course of bleo treatment and not just at day 22.

- Thank you for this observation. The sentence was modified to “the expression of CIRP was higher in patients with acute exacerbation of PF than in patients with stable PF” (lines 38-39).

3. Clarify in methods what was harvested for qPCR, mouse lung and what lobes since portions were taken for histology and hydroxyproline assays?

- The left lung was fixed in formalin for histological assessment and the right lung was used for qPCR (superior and middle lobes) and hydroxyproline content measurement (inferior and post-caval lobes). This information has been added to the Materials and Methods section (lines 102-104).

4. Did you notice any heterogeneity in the histological sections of mouse lung tissue? Could this variability account for the wide spread in your WT Bleo mice? Could you speak to why you might see such variability in these samples? Also was the scoring of the fibrotic lung tissue blinded?

- The histological assessment was performed in a blinded fashion; this information has been added to the Materials and Methods section (lines 153-154). We did not observe histological heterogeneity that might explain the overall variability of the averaged Hübner-modified Ashcroft scores.

5. I would be curious to see if preemptive treatment or co-treatment with C23 with the bleo could further reduce or prevent the development of the fibrotic phenotype observed.

- We would like to thank the reviewer for this insightful comment. At the design stage of this project, we did consider an arm of pre-treatment with C23. However, our group is more interested in new treatments for PF than in its etiology. As such, since pre-treatment is not a realistic therapeutic approach for PF patients, we decided not to proceed with this arm in the current project. Nonetheless, we will consider the Reviewer’s suggestion in our future studies. 

6. I recommend including a section in the discussion about the application of C23 in patient treatment, potential issues, clinical use etc. to frame what is known and still needs to be understood.

- Thank you very for that excellent recommendation. We have now included the sequence and source of C32 (Materials and Methods, line 106) and included the knowledge gaps that need to be addressed in reference to C23 (Discussion, lines 383-387).

---

## [Decision Letter · Decision Letter 1]

16 Mar 2022

The Role of eCIRP in Bleomycin-Induced Pulmonary Fibrosis in Mice

PONE-D-21-22312R1

Dear Dr. Brenner,

We’re pleased to inform you that your manuscript has been judged scientifically suitable for publication and will be formally accepted for publication once it meets all outstanding technical requirements.

Kind regards,

Manjula Karpurapu

Academic Editor

PLOS ONE

Reviewers' comments:

Reviewer's Responses to Questions

**Comments to the Author**

1. If the authors have adequately addressed your comments raised in a previous round of review and you feel that this manuscript is now acceptable for publication, you may indicate that here to bypass the “Comments to the Author” section, enter your conflict of interest statement in the “Confidential to Editor” section, and submit your "Accept" recommendation.

Reviewer #1: All comments have been addressed

Reviewer #2: All comments have been addressed

2. Is the manuscript technically sound, and do the data support the conclusions?

Reviewer #1: Yes

Reviewer #2: Yes

3. Has the statistical analysis been performed appropriately and rigorously? 

Reviewer #1: Yes

Reviewer #2: Yes

4. Have the authors made all data underlying the findings in their manuscript fully available?

Reviewer #1: Yes

Reviewer #2: Yes

5. Is the manuscript presented in an intelligible fashion and written in standard English?

Reviewer #1: Yes

Reviewer #2: Yes

6. Review Comments to the Author

Reviewer #1: (No Response)

Reviewer #2: The authors have addressed my comments where they could and answered my questions where they could not. This study leaves a lot to be investigated but the authors do not over reach on their claims or conclusions of their studies so far.

7. PLOS authors have the option to publish the peer review history of their article (what does this mean?). If published, this will include your full peer review and any attached files.

Reviewer #1: No

Reviewer #2: No

---

## [Editor Report · Acceptance letter]

25 Mar 2022

PONE-D-21-22312R1 

The Role of eCIRP in Bleomycin-Induced Pulmonary Fibrosis in Mice 

Dear Dr. Brenner:

I'm pleased to inform you that your manuscript has been deemed suitable for publication in PLOS ONE. Congratulations! Your manuscript is now with our production department. 

Kind regards, 

on behalf of

Dr. Manjula Karpurapu 

Academic Editor

PLOS ONE